# Environmental Gamma Dose Rate Measurements using CZT Detectors

Sebastian Kreutzer[1, *], Loïc Martin[2], Didier Miallier[3], and Norbert Mercier[4]

[1]Institute of Geography, Heidelberg University, Im Neuenheimer Feld 348, 69120 Heidelberg, Germany
[2]Innsbruck Quaternary Research Group, University of Innsbruck, Innrain 52, 6020 Innsbruck, Austria
[3]Laboratoire de Physique de Clermont-Ferrand, Université Clermont Auvergne, Campus des Cézeaux, 24, avenue des Landais BP 80026, 63171 Aubière cedex, France
[4]Archéosciences Bordeaux, CNRS-Université Bordeaux Montaigne, Maison de l'Archéologie, Esplanade des Antilles, 33607 Pessac cedex, France

**Correspondence:** Sebastian Kreutzer (sebastian.kreutzer@uni-heidelberg.de)

**Abstract.** The accurate and precise determination of the environmental dose rate is pivotal in every trapped-charge dating study. The environmental $\gamma$-dose rate component can be determined from radionuclide concentrations using conversion factors or directly measured in situ with passive or active detectors. In-field measurements with an active detector are usually inexpensive and straightforward to achieve with adequate equipment and calibration. However, despite the rather widespread use of portable NaI or LaBr$_3$ scintillator detectors, there is a lack of research on the performance and practicality of portable alternative detectors in dating studies, particularly in light of newer developments in the semi-conductor industry. Here, we present our experience with two small portable semi-conductor detectors housing Cadmium Zinc Telluride (CZT) crystals. Given their small volume and low power consumption, we argue they present attractive alternatives for $\gamma$-dose rate measurements in dating studies. Despite high relative detection efficiency, their small volume may pose different challenges, resulting in impractical measurements in routine studies and, therefore need investigation. In our study, we simulated the particle interaction of the CZT crystal with *GEANT4* in different sediment matrices to quantify the energy threshold in the spectrum above which the count/energy-count rate correlates with the environmental gamma dose-rate irrespective of the origin of the gamma-photons. We compared these findings with experimentally derived cumulative spectra and dose-rate calibration curves constructed from reference sites in France and Germany, which yielded unrealistically low threshold values likely due to the limited variability of the investigated sites. We additionally report negligible equipment background and required minimal measurement time of only $20\,\mathrm{min}$ in typical environments. Cross-checking our calibration on a homogenous loess deposit near Heidelberg confirmed the setting and assumed performance through a nearly identical $\gamma$-dose rate of $1107 \pm 65\,\mathrm{\mu Gy\,a^{-1}}$ (CZT) to $1105 \pm 11\,\mathrm{\mu Gy\,a^{-1}}$ (laboratory). The outcome of our study gives credit to our threshold definition. It validates the similarity of the two investigated probes, which may make it straightforward for other laboratories to implement the technique effortlessly. Finally, the implementation of CZT detectors has the potential to streamline fieldwork and enhance accuracy and precision of trapped-charge dating-based-chronologies.

# 1 Introduction

Assessing the effective environmental dose rate is indeed crucial for accurate and precise ages in luminescence and electron-spin-resonance (short: trapped-charge) dating studies. The dose rate plays a vital role in the age equation as it is a significant
factor in determining the amount of radiation absorbed over time.

Field procedures typically involve sampling sufficient bulk material around the sampling site. The material is then analysed to quantify the natural radionuclide concentrations (such as U, Th, and K concentrations) as major contributors to the environmental radiation. If collecting sufficient material is not feasible around the sampling site, e.g., in archaeological excavations, or drilled cores, the required material can be carefully separated from the to-be-dated material combined with in situ
measurements.

Strategies to ensure a good environmental dose rate estimation ideally include laboratory and field measurements. Standard analytical methods in the laboratory involve $\alpha$-/$\beta-$counting, $\gamma$-ray spectrometry (e.g., Aitken, 1985; Zöller and Pernicka, 1989; Hutton and Prescott, 1992; Preusser and Kasper, 2001; Godfrey-Smith et al., 2005; Mauz et al., 2021; Kolb et al., 2022) and element analytical methods such as inductively coupled plasma mass spectrometry (ICP-MS) (e.g., Preusser and Kasper, 2001)
or a combination of those methods. Activity or element concentrations are then converted for each type of radiation ($\alpha$-, $\beta$, $\gamma$-radiation) using dose-rate conversion factors (latest update: Cresswell et al., 2018).

Dose-rate ($\gamma$- and, more challenging, $\beta$-) components can be measured in the field at the sampling position using passive dosimeters (e.g., Hutton and Prescott, 1992; Kalchgruber et al., 2003; Kalchgruber and Wagner, 2006; Richter et al., 2010; Kreutzer et al., 2018b) stored over a couple of weeks to months or with active detectors (usually $\gamma$-ray probes) (e.g., Murray
et al., 1978; Mercier and Falguères, 2007; Guérin and Mercier, 2011; Arnold et al., 2012; Bu et al., 2021; Martin et al., 2024) enabling nearly instant dose rate estimates.

Regardless of the preferred method and type of detector, active or passive, in-field measurements are indicated when sampling suggests a heterogeneous distribution of radionuclides or complex geometries (e.g., the close succession of very different sediment layers, gravels/rocks in the profile). The field dose-rates can later be compared to laboratory results based on the
radionuclide concentrations. Ideally, the obtained numbers statistically agree, or the discrepancy gives further insight into the site's matrix composition. Active detectors can be paired with a portable luminescence reader (e.g., Sanderson and Murphy, 2010) to profile the stratigraphy and determine relative chronologies. A rule of thumb would approximate the $\gamma$-dose component as about 28% to 36% of the total dose rate (numbers derived from the ChronoLoess database by Bosq et al., 2023; alternatively, see estimates in Aitken, 1985). These numbers underpin the importance of the $\gamma$-dose rate contribution and its
significance in estimating accurate trapped-charge ages.

On the flip side, the usually short measurement durations, compared to the expected age of the sediment, have the disadvantage that long-term changes in the water content are not reflected. Contrary, passive dosimeters would register at least seasonal variations if stored over months in the field. Both (passive and active) do not register potential radioactive disequilibria. Furthermore, depending on the size of the detector probe a rather large hole with a depth of at least ca $30\,\mathrm{cm}$ is required for the
measurement. Such a hole is sometimes difficult to dig, not always possible (samples from a drilled core) or not favoured

given the setting (e.g., archaeological excavation). Here focussing on active detectors, additional everyday challenges involve equipment calibration and handling usually proprietary hardware such as cables or multi-channel analysers that are costly to repair or even unavailable after they have been phased out by the manufacturer. Last, the equipment can be bulky, especially for large detectors (up to $3 \times 3\,\text{in}$ for a portable NaI probe) and, given first-hand experience, the equipment is prone to preferred inspection during air travel and require an export port license due to being dual use and cannot be brought into every country.

In summary, while in-field measurements with active detectors do present certain challenges, their benefits are still considerable. They provide valuable, real-time data at a relatively low cost, significantly improving the accuracy of dating studies. As a result, their routine use seems advisable.

In the following, we will test two commercially available portable Cadmium Zinc Telluride (CZT) detectors for in situ $\gamma$-ray measurements. The detectors are small and highly portable, and we assume that they can pose an alternative to systems using larger NaI or LaBr$_3$ probes in trapped-charge dating applications. Next, we will begin outlining the technical specifications and advantages of the CZT systems. We will then detail the required calibration methods and explore the performance and dose-response characteristics of the detectors through simulations and measurements in different natural sites with well-known radionuclide concentrations. Finally, we will test the calibrated systems in a loess deposit near Heidelberg and discuss the results.

In this contribution, we focus exclusively on the "threshold" technique (Løvborg and Kirkegaard, 1974, further details below) for measuring environmental $\gamma$-dose rates ($\dot{D}_\gamma$ in $\mu\text{Gy}\,\text{a}^{-1}$). Unlike the "window" method (three windows each for U, Th, and K), which compares the area under a specific $\gamma$-peak in a sample with unknown composition to the area of a $\gamma$-peak in a sample with known radionuclide composition, the threshold technique integrates the entire spectrum above a set threshold. The threshold approach provides a direct measure of $\dot{D}_\gamma$ in $\mu\text{Gy}\,\text{a}^{-1}$ rather than a radionuclide concentration for natural environments.

## 2  Material and methods

### 2.1  Brief background $\gamma$-detectors

Measuring $\gamma$-rays translates to observing the interaction of ($\gamma$-) photons with matter by quantifying the production of secondary charged particles. Suitable are scintillation detectors such as NaI(Tl) or LaBr$_3$, collecting light caused by the interaction of the $\gamma$-photons with the detector material. Alternatively, semiconductor-based detectors (e.g., high-purity Ge) combined with a suitable electronics, record the amount of produced secondary-hole pairs (e.g., Gilmore, 2008).

To measure $\gamma$-rays outside a laboratory, for instance, in trapped-charge dating studies, portable detectors that can be operated at room temperature are preferred. This usually favours scintillation detectors using NaI(Tl) or LaBr$_3$ with typical probes ranging from $1.5 \times 1.5\,\text{in}$ to $3 \times 3\,\text{in}$ over HPGe semiconductor-based detectors that require operation at liquid nitrogen temperature due to small band-gap of the crystal. Cadmium Zinc Telluride (CdZnTe; short: CZT) were proposed as promising alternatives with better $\gamma$-ray absorption performance and operational at room temperature. However, the production process

is more challenging (e.g., Gilmore, 2008) and such detectors were not an option considered in the context of trapped-charge geochronology; yet.

Since the 1990s, the development of CZT semiconductor detectors progressed considerably in their applicability as $\gamma$-ray detectors (for reviews, see Scheiber and Chambron, 1992; Verger et al., 1997; Limousin, 2003; Alam et al., 2021). They offer a small volume and operate at ambient temperature by collecting charges created by the interaction of ionising radiation with a high relative efficiency for photoelectric interaction (atomic numbers Cd: 48, Te: 52; density crystal ca $5.8\,\mathrm{g\,cm^{-3}}$) (Limousin, 2003; Alam et al., 2021). Although this is less important in our case, they provide an energy resolution comparable to or better than LaBr$_3$ and considerably higher than NaI(Tl) (Alexiev et al., 2008) probes. Also considering the small volume available for detection, their absolute efficiency remains lower than that of larger detectors. This feature, combined with a low energy consumption, renders this detector type particularly appealing for our application.

## 2.2 Equipment

For our experiments, we used two systems from Kromek (https://www.kromek.com/; last access: 2024-08-17) with CZT detectors. (1) RayMon10® (henceforth: RayMon GR1) and (2) GR1+® (henceforth: GR1)[1] (Fig. 1). Both systems include a similar $10 \times 10 \times 10\,\mathrm{mm^3}$ GR1 CZT detector connected to a 4096 energy/channels analyser. The detection ranges from $30\,\mathrm{keV}$ to $3\,\mathrm{MeV}$ with an energy resolution of around 2.5% FWHM at $662\,\mathrm{keV}$. The RayMon GR1 was delivered with a handheld touch-screen device running Microsoft Windows 10® and comes housed. The probe communicates with the handheld device via a Universal Serial Bus (USB) Type A connector. The battery lasts around eight to ten hours, depending on the display brightness setting. Although much smaller in housing size (GR1: $25\,\mathrm{mm} \times 25\,\mathrm{mm} \times 63\,\mathrm{mm}$, $60\,\mathrm{g}$; compared to RayMon GR1: $42\,\mathrm{mm} \times 35\,\mathrm{mm} \times 180\,\mathrm{mm}$, $176\,\mathrm{g}$), the GR1 contains a similar CZT detector. It has a Mini-A USB port that can be attached to any standard computer given a suitable cable and operated using the software K-Spect® that can be downloaded free of charge from the manufacturer. The GR1 consumes only $250\,\mathrm{mW}$ and is hence operational as long as the battery of the connected computer lasts. For more information, we refer to the manufacturer's website.

Because the Mini-USB port of the GR1® seemed fragile, and we were not sure about the sealing of the housing against moisture, we designed a 3D-printed, rubber-sealed strain relief mount (Fig. 1) and attached it to the detector housing. The strain relief enables safe retrieval of the detector, and the simple plastic bag wrapping keeps dirt and moisture away during field operations. The adapter was designed by the Scientific Workshop Service of Heidelberg University and we share the print-ready files under CC BY-NC 4.0 licence conditions on Zenodo as a supplement to this article.

## 2.3 Calibration methods

We aim to use the detectors in routine dating applications to determine $\dot{D}_\gamma$ in $\mathrm{\mu Gy\,a^{-1}}$. This requires three separate experiments in given order to set up each device: (1) Channel/energy calibration, (2) energy threshold definition, and (3) a calibration curve modelling counts against the environmental dose rate.

---

[1]The plus indicates a slightly higher energy resolution compared to the "non-plus" GR1 version.

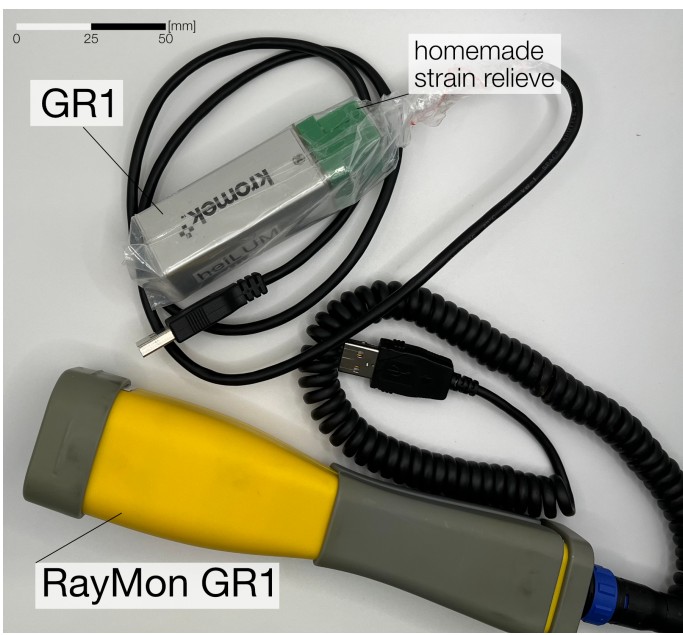

**Figure 1.** Kromek detectors used for our measurements (shown is the probe without the handheld tablet PC for the RayMon10®). Both probes house a similar CZT detector. We wrapped the GR1® in a standard plastic bag and attached a home made strain relief to the GR1® to enable easier operation and retraction of the detector in the field.

The dead time was insignificant during all the experiments presented in this manuscript. The dead time is the difference between real-time and live time, which equals the time when the detector did not register new counts. The longest dead time of both detectors was 3.6 s with the highest relative dead time amounting to only 0.1 %.

### 2.3.1 Channel/energy calibration

The channel/energy calibration (Sec. 3.1) assigns energy values in keV to the, in our case, 4096 channels. The calibration makes it easier to interpret the $\gamma$-ray spectrum, enables a comparison of spectra, and accounts for shifts in the spectrum that may occur due to, for instance, changed environmental conditions.

Both detectors used here were delivered with a test and inspection sheet documenting measurements against $^{241}$Am ($\gamma$-line at 59.5 keV) and $^{137}$Cs ($\gamma$-line at 662 keV). The results are nearly identical for both detectors with an offset of ca 2 channels/keV between the GR1 and the RayMon GR1.

For the channel/energy calibrations, where only the peak position matters, we used two $\gamma$-standards available in Heidelberg closely arranged around the detector for two measurements over 3600 s. One source is a home-made uranium standard (U concentration: 1.02%) and the other an Amersham EB 165 mixed radionuclide standard with $^{241}$Am and $^{137}$Cs. The Amersham standard also contains other shorter-lived radionuclides; however, given the age of the standard (>30 years), we do not expect to observe significant counts above background within the chosen measurement time.

### 2.3.2 Energy threshold determination

The energy-threshold definition (Sec. 3.5) determines the threshold in the spectrum above which $\dot{D}_\gamma$ is seemingly independent of the origin of the absorbed $\gamma$-photons (see Løvborg and Kirkegaard, 1974, for details). In other words, the integrated spectrum above the threshold is used to derive $\dot{D}_\gamma$. Guérin and Mercier (2011) distinguished two different thresholds techniques for *integrating* the spectrum. The "count" and the "energy" threshold (integration technique). The *count* threshold adds all counts above a certain threshold ($\eta$) whereas the *energy* threshold integrates the deposited energy above $\eta$. Assuming that $S_i$ is the

signal registered either as absolute counts per channel or count rate per channel ($s^{-1}$) in the $i^{\text{th}}$ channel of the spectrum, $E_i$ in keV is the energy associated with a certain channel. The relationship between the environmental $\gamma$-dose rate and integrated value above the threshold for an energy/channel calibrated spectrum becomes, in case of the counting threshold technique:

$$\dot{D}_\gamma \sim \Sigma_{i:=\eta}^N S_i \tag{1}$$

and it reads

$$\dot{D}_\gamma \sim \Sigma_{i:=\eta}^N S_i \times E_i \tag{2}$$

for the energy threshold integration technique. Guérin and Mercier (2011) found $\eta$ slightly lower for the latter technique, resulting in a larger proportion of the spectrum usable, which lowers the statistical uncertainty. Although related, the two threshold integration techniques must be distinguished from quantifying $\eta$, i.e. finding the energy above which $\dot{D}_\gamma$ is a function of the integrated counts (Løvborg and Kirkegaard, 1974), regardless of the integration technique. To determine $\eta$, one can

perform energy-matter interaction simulations (e.g., Guérin and Mercier, 2011) or measure the $\gamma$-ray spectra of "pure" emitters of known U, Th, K concentrations (Mercier and Falguères, 2007; Rhodes and Schwenninger, 2007; Duval and Arnold, 2013).

For our contribution, we modelled the threshold (henceforth: $\eta_{sim}$ in keV) with *GEANT4* (Agostinelli et al., 2003) using three different sediment matrices: (1) a pure $SiO_2$ matrix, (2) a brick-like matrix ($SiO_2$: 66 %, $Al_2O_3$: 18 %, $Fe_2O_3$: 6 %), and (3) a calcite rich sediment ($CaCO_3$: 60 %, $SiO_2$: 40 %). We set the matrix densities to $1.8\,\mathrm{g\,cm^{-3}}$, and added no water (dry

matrix). The simulation geometry represents the RayMon10® probe according to the available documentation provided by the manufacturer (Fig. 2). Dimensions and material of the prove were provided by the manufacturer. The probe was placed at the centre of the $(1.635 \times 1.635 \times 1.67)\mathrm{m^{-1}}$ sediment box. In this geometry the probe is surrounded by at least $80\,\mathrm{cm}$ of sediment in every direction, ensuring an infinite $\gamma$-radiation matrix around it (>99% of the infinite matrix $\gamma$-dose rate). The $\gamma$-ray emissions were simulated from each matrix using *GEANT4* electromagnetic physics from the `G4EmPenelopePhysics` constructor

(Ivanchenko et al., 2011), which is based on the 2008 version of the *PENELOPE* Monte Carlo code for low energy particles (Baró et al., 1995). This *GEANT4* physics was successfully already tested for simulating $\gamma$-photons from natural radionuclides in Guérin and Mercier (2011), Guérin and Mercier (2012), Martin et al. (2015).

The $\gamma$-spectra of $^{40}$K, the U-series (in secular equilibrium) and the Th-series (in secular equilibrium) were built from the data of the ENSDF database as of June 2014 (http://www.nndc.bnl.gov/ensarchivals/; last access: 2024-09-15) to independently

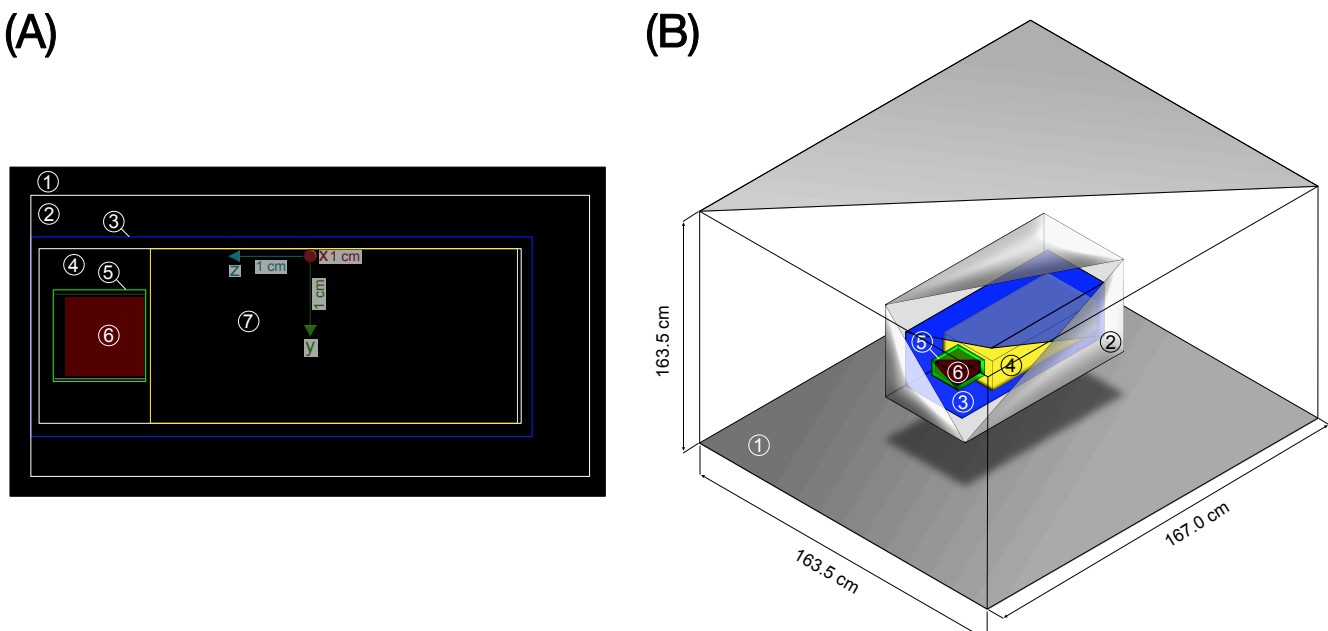

**Figure 2.** Geometry of the simulated CZT detector and its placement in the sediment matrix box. (A) Image from *GEANT4* visualization only showing the detector part. (B) Isometric plot showing the placement in the sediment matrix box. Both graphics to not scale. Legend: (1) sediment matrix; (2) rubber coating; (3) aluminum shielding; (4) air; (5) plastic support; (6) CZT crystal; (7) electronic components. Technical information used for building the geometry were kindly provided by the Kromek Group plc. The dimensions of the sediment matrix box are $(1.635 \times 1.635 \times 1.67)\,\mathrm{m}^3$. The production cut for the simulation was $0.5\,\mathrm{keV}$. For full details on dimensions and composition we refer to the *GEANT4* code available on Zenodo along with this contribution.

simulate $1\,\mathrm{Gy}$ of $\gamma$-dose in the matrices. In the simulation, we recorded the energy of each $\gamma$-interaction with the CZT crystal, and the spectra of counts per energy channel were built for each matrix and each $\gamma$-emission spectra. These "measured" spectra obtained by simulation were then used to create the curve of counts/deposited energy above the energy thresholds ranging from $0\,\mathrm{keV}$ to $1000\,\mathrm{keV}$. We then compared the standard deviation between the $^{40}\mathrm{K}$, U-series and Th-series curves of counts above the threshold to quantify the optimal threshold for which the number of counts/energy above is proportional to the dose
rate and mostly independent of the natural radionuclide composition. We did not consider dead times because we assume that, during real couting, this phenomenon has a low impact on determining the count/energy threshold.

We compare these findings with measurements at five calibration sites (Fig. 3) to derive $\eta_{exp}$. Four sites are located in France, three in the vicinity of Clermont-Ferrand (France) (Miallier et al., 2009), and one is a home-made brick block located in the cellar of the Archéosciences Bordeaux laboratory (Richter et al., 2010). Another site is a granite block (FLOSSI) located at the
Max Planck Institute for Nuclear Physics (Heidelberg, Germany). The granite block was donated by the Granitwerke Leonhard Jakob KG to the Forschungsstelle Archäometrie (Günther A. Wagner) in 1991 for the purpose of having a reference site for

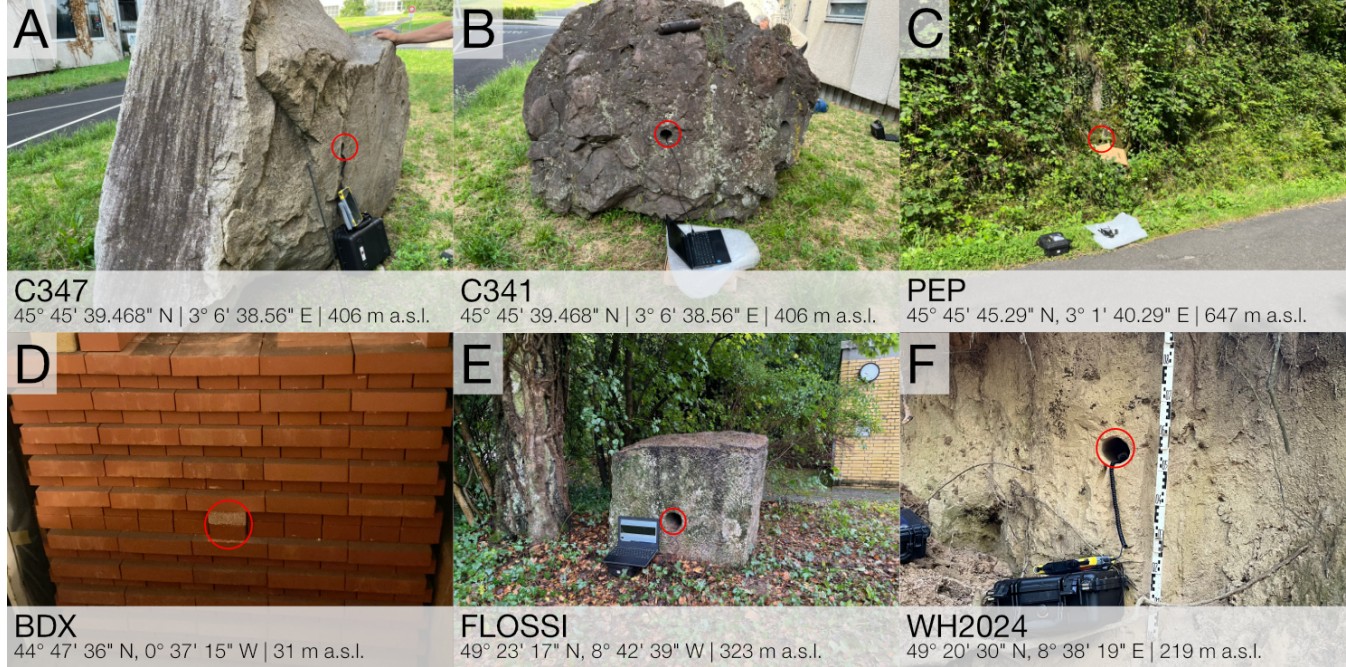

**Figure 3.** Photos of all natural sites mesaured in this study. A-E are calibration sites with known radionuclide composition, (F) is a loess deposit at the Weiße-Hohl near Heidelberg (Germany) used to cross-check the equipment calibrations. Details to A-D can be found in Miallier et al. (2009) and Richter et al. (2010), details to E and F are provided in the maintext. The red circles mark the measurement positions (holes) for the probe. The sites BDX (D) and FLOSSI (E) are located in areas with restricted access. (D) in the basement of the Archéosciences Bordeaux laboratory at the Université Bordeaux Montaigne in Pessac (France) and (E) at the Max Planck Institute for Nuclear Physics in Heidelberg (Germany).

calibrating $\gamma$-ray spectrometers. The radioelement concentration of the block was analysed with neutron activation analysis, atomic absorption and high-resolution $\gamma$-ray spectrometry as part of the work by Rieser (1991). Although the information from this analysis was later used by others (e.g., Hossain et al., 2002; Kalchgruber, 2002) the values were never formally published. We therefore added the CSV file with the values from Rieser (1991) to our Zenodo dataset (Kreutzer et al., 2024).


In general, the investigated sites (Fig. 3A-E) have a well-known radionuclide composition from which $\dot{D}_\gamma$ can be calculated to construct $\gamma$-dose rate calibration curves using the two threshold integration techniques to re-evaluate $\eta$ as the value where the mean square of residuals from the model reaches the lowest value. The underlying assumption of this approach is that if the threshold was set correctly, the regression line should exhibit the best fit as a non-ideal threshold should increase the residuals due to a poor fit. Insufficient model adaptation is caused by poor counting statistics (threshold too large) or in situations where


the prerequisite of the technique that $\dot{D}_\gamma$ is independent of the origin of the $\gamma$-photons is not fulfilled (threshold too low). We will compare those values with the experimental approach of deriving the threshold, except that we do not have access to sites with pure radionuclide concentrations but will use measurements from sites Fig. 3A-E instead.

**Table 1.** Known $\gamma$-dose rates from reference sites. The dataset listed here, except for FLOSSI, ships with the R package 'gamma'. In the original dataset, 'BDX' is termed 'BRIQUE', which is the brick block in the Archéosciences Bordeaux laboratory; for clarity, we relabeled it to BDX for our analysis. The values for FLOSSI represent the central values and standard error (Galbraith and Roberts, 2012) of all respective analyses given in Rieser (1991). We list the results calculated with the conversion factors by Cresswell et al. (2018). Please note that data in Miallier et al. (2009) are given as total dose rate, including the cosmic-dose rate contribution. Here we can neglect the cosmic dose rate contribution as we cut the spectra at 2800 keV.

| SITE | NATURE | U | $\sigma_U$ | Th | $\sigma_{Th}$ | K | $\sigma_K$ | $\dot{D}_\gamma$ | $\sigma_{\dot{D}_\gamma}$ |
|---|---|---|---|---|---|---|---|---|---|
| BDX | ceramic | 4.1 | 0.1 | 13.7 | 0.4 | 3.5 | 0.1 | 1997.1 | 37.7 |
| C341 | trachybasalt | 1.8 | 0.0 | 6.4 | 0.4 | 1.4 | 0.0 | 855.2 | 21.8 |
| C347 | granite | 2.8 | 0.1 | 4.7 | 0.1 | 3.5 | 0.1 | 1425.5 | 27.4 |
| FLOSSI | granite | 19.2 | 0.6 | 13.4 | 0.4 | 4.1 | 0.1 | 3797.4 | 92.5 |
| PEP | granite | 6.0 | 0.2 | 19.0 | 2.0 | 3.8 | 0.2 | 2554.3 | 112.7 |

*Note:*

U, Th concentrations in $\mu g\,g^{-1}$, K in % | dose rates in $\mu Gy\,a^{-1}$.

### 2.3.3 Dose-rate calibration curves

The dose-rate calibration curve (Sec. 3.6) correlates the integrated (count and energy integration technique) signal with $\dot{D}_\gamma$ from the reference sites (Table 1), i.e. the response of the detector to natural $\gamma$-radiation. If established, it allows us to derive an accurate estimate of $\dot{D}_\gamma$ from a natural site with unknown radionuclide composition. As pointed out by Guérin and Mercier (2011), the water content will not affect the counting rate significantly, and the established value should be applicable to sites usually probed in trapped-charge dating applications.

The $\dot{D}_\gamma$ values in Table 1 differ from the values reported in Miallier et al. (2009) after we recalculated them using the conversion factors compiled by Cresswell et al. (2018). Values recalculated for other conversion factors can be found in the dataset `clermont_2024` contained in the R `'gamma'` package (> v1.1.0).

Please note that for establishing the calibration curves we assumed "infinite matrix" conditions that enabled us to convert the radio-nuclide concentrations into dose rates (e.g., Guérin et al., 2012, for a critical review of this concept).

### 2.4 Radionuclide determination cross-check

To validate our calibration and post-processing procedure, we recorded natural $\gamma$-spectra at the Weiße Hohl (WH2024). The site is a gully of anthropogenic origin that cut into the famous last-glacial aeolian deposits near Nussloch (Germany) (e.g., Antoine et al., 2001). Today, the gully is part of a hiking trail in the area and hence easily accessible. The Nussloch loess deposits are well-investigated through numerous studies, and the expected $\dot{D}_\gamma$ at Weiße Hohl was about $1\,Gy\,ka^{-1}$ (Rieser, 1991). What made the measurements at this particular site interesting was that loess is typically subject to past climate and chronology studies using trapped-charge dating methods and reflect an often encountered use case.

We recorded two spectra over $20\,\text{min}$ with both detectors in a $32\,\text{cm}$ deep hole and sampled about $120\,\text{g}$ of material for subsequent radionuclide and gravimetric water content quantification. We further extracted two subsamples for radionuclide concentration analyses in Heidelberg and Bordeaux. In Heidelberg, we employed a $\mu$Dose (Tudyka et al., 2018; Kolb et al., 2022) and a $\mu$Dose+ (Tudyka et al., 2024) system on the same $3\,\text{g}$ subsample. The sample was measured more than two days on each system. On another $83.3\,\text{g}$, we performed high–resolution $\gamma$-ray spectrometry measurements in Bordeaux (Guibert and Schvoerer, 1991). To compare the $\dot{D}_\gamma$ calculated from the radionuclide concentrations using the conversion factors by Cresswell et al. (2018). We corrected the $\dot{D}_\gamma$ measured with GR1 and RayMon GR1 for the field water content (Aitken, 1985). As for the calibration measurements, we assumed "infinite matrix" conditions and approximated a $4\pi$ geometry.

## 2.5 Data and data processing

We used *GEANT4* (Agostinelli et al., 2003) for the threshold modelling and processed our data with R (R Core Team, 2024) and the packages `'gamma'` (Lebrun et al., 2020; Frerebeau et al., 2024) and `'ggplot2'` (Wickham, 2016). The two investigated Kromek measurement systems provide export functionality for various data formats. We opted for the ASCII format `.spe` and added support in the function `gamma::read()` to the package `'gamma'` (> v1.1.0) for this study (Frerebeau et al., 2024). Except additions detailed below, our workflow uses the analysis functions of the `'gamma'` package and follows the suggestions by Lebrun et al. (2020) and the tutorials that come with the `'gamma'` (>v1.1.0) R package. This also includes the steps to determine the dose-rate response curve. For clarity, it should be mentioned that the `'gamma'` package internally uses the function `IsoplotR::york()` (Vermeesch, 2018) to implement a regression analysis with correlated errors of xy-values that have individual uncertainties (York et al., 2004).

To ensure that the figures have colour-blind-friendly colours, we used the R package `'khroma'` (Frerebeau, 2024) and the manuscript was prepared with `'rticles'` (Allaire et al., 2024). A shortened version of the R code used for all the calculations, data, and calibration output are available on Zenodo (Kreutzer et al., 2024) under CC BY 4.0 licence conditions in accordance with common data-sharing guidelines.

# 3 Results

## 3.1 Energy calibration

Figure 4 shows the spectrum plot of GR1 measured over $1\,\text{h}$. We placed our $\gamma$-standards with known composition in front of the detector. The dashed lines marked the $\gamma$-lines used for the channel/energy calibration. The inset draws the energy/channel-calibration curve applied subsequently to all analysed spectra. We did not apply low-level discrimination and recorded raw count values for the measurements, i.e. the count rate was calculated in the post-processing. We have chosen the measurement time to achieve a good counting statistic.

Given the nuclide composition, we expected to see typical $\gamma$-lines present in the $^{238}$U decay chain on top of $^{241}$Am and $^{137}$Cs. The manufacturer also used the latter two nuclides before delivery to test the CZT detectors' performance and hence

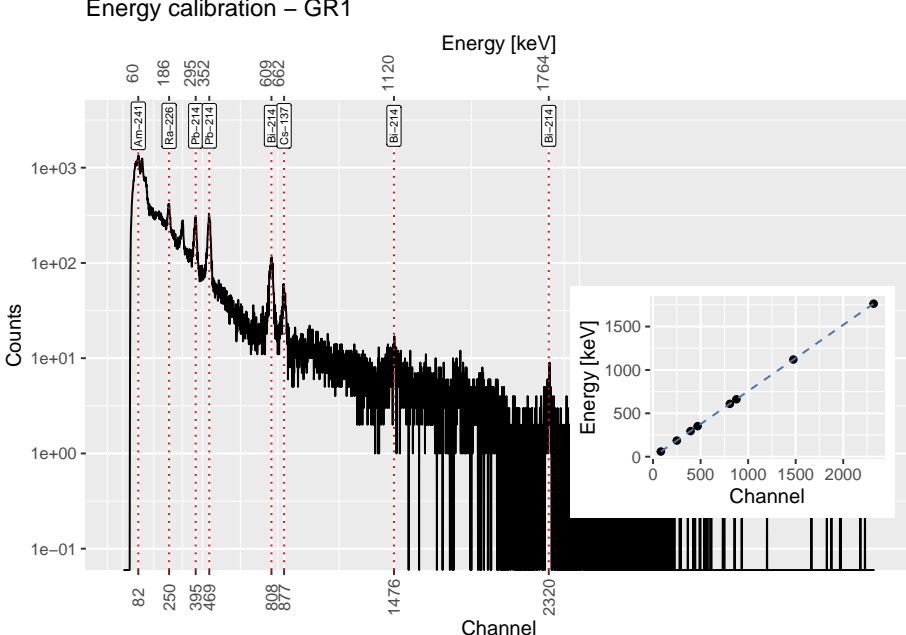

**Figure 4.** Energy calibration results for detector GR1. The main plot shows the raw spectrum with known $\gamma$-lines marked with dashed lines. The inset displays the calibration curve applied to all subsequently shown spectra. Peak positions were found to be similar for GR1 and RayMon GR1.

they provide a good reference for a cross-check. We manually identified eight $\gamma$-lines in our spectrum and assigned the results to the imported spectra with the `gamma::energy_calibrate()` function. To ease the peak identification, we started with

the $^{241}$Am and $^{137}$Cs $\gamma$-lines for which we have channel-to-energy references determined by the manufacturer. For instance, the manufacturer specifies to find the $^{241}$Am peak @ 59.5 keV in channel number 80 ($\pm 10\,\%$) and the $^{137}$Cs peak at 662 keV at 880 ($\pm\,1\,\%$). Our calibration confirmed those values with channel number 82 for $^{241}$Am 59.5 keV and channel number 877 for $^{137}$Cs 662 keV.

The same calibration was performed with the RayMon GR1 detector but with its measurement time reduced to 900 s as a

cross-check. According to test data shared by the manufacturer on request (personal communication via e-mail, 2024-09-27), peak areas do not differ by more than 5% to 10% if the equipment is operated within the specified range (0-40°). In our case, we found that the peak positions of RayMon GR1 were virtually identical to GR1 also under different temperature conditions (ambient temperatures at Clermont Ferrand: ca 28°, at FLOSSI: ca 18°; data not shown). Hence, for simplicity, we applied the GR1 channel/energy calibration to all measured spectra, and all spectra shown subsequently are energy/channel-calibrated.

## 3.2   Background measurements

To investigate the detector's counting background, we placed the GR1 for ca 5 h (18060 s) in a lead housing inside a low-level background environment at the PRISNA facility (Plateforme Régionale Interdisciplinaire de Spectroscopie Nucléaire en

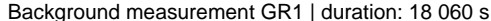

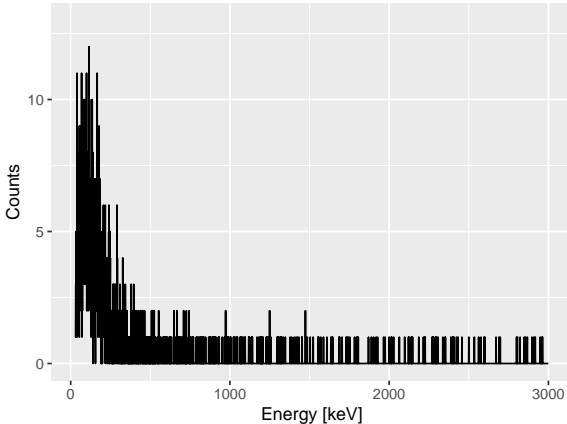

**Figure 5.** Background measurements with GR1 in the lead castle for more than 5 h. The system background is negligible compared to typical measurements in the field.

Aquitaine) near Bordeaux. The facility is a scientific platform used for low-level $\gamma$-ray spectroscopy experiments. Figure 5 illustrates that the system background is insignificant compared to typical environmental situations. The average count rate of
the sum spectrum amounts to only $0.1\,\mathrm{s}^{-1}$. Given the similarity of both detectors, we applied the same background subtraction to results from both detectors.

### 3.3 Energy-calibrated raw spectra

Figure 6 displays all energy calibrated $\gamma$-ray spectra measured at the sites at Clermont-Ferrand, Bordeaux, Heidelberg and PRISNA. We show count rates instead of absolute count values to account for the different measurement time. The shortest
live time was 1200 s (PEP) and the longest 18059 s (PRI). All spectra in Fig. 6(A) and (B) are scaled and colour-coded similarly for better comparison. Site PRI (the background measurement) was only measured with the GR1.

Visible in both spectra is a dominant Compton continuum rather than distinguishable photo peaks. This observation is not surprising given the short measurement time, the low abundance of the radionuclides (e.g., Miallier et al., 2009, for the Clermont-Ferrand sites) and, of course, the relatively low absolute efficiency for the small CZT crystal. It is reassuring that all
comparable raw spectra appear very similar in intensity, position and shape, except for the C341 spectra.

The spectra recorded in site C341 (a basaltic rock) appear to show only half of the counts measured with detector GR1 compared to the RayMon GR1 detector. This discrepancy is because GR1 was controlled via an external mobile computer that went unexpectedly into sleep mode. After reactivating the computer, the software seemed to have continued counting. However, post-processing revealed that it had stopped registering $\gamma$-photons. In other words, the difference between the two
readings (GR1 vs RayMon GR1) for C341 is a technical error, and hence, we discarded the spectrum C341 measured with GR1 for subsequent analysis. This error can be avoided easily but we kept it in the manuscript to share our experience.

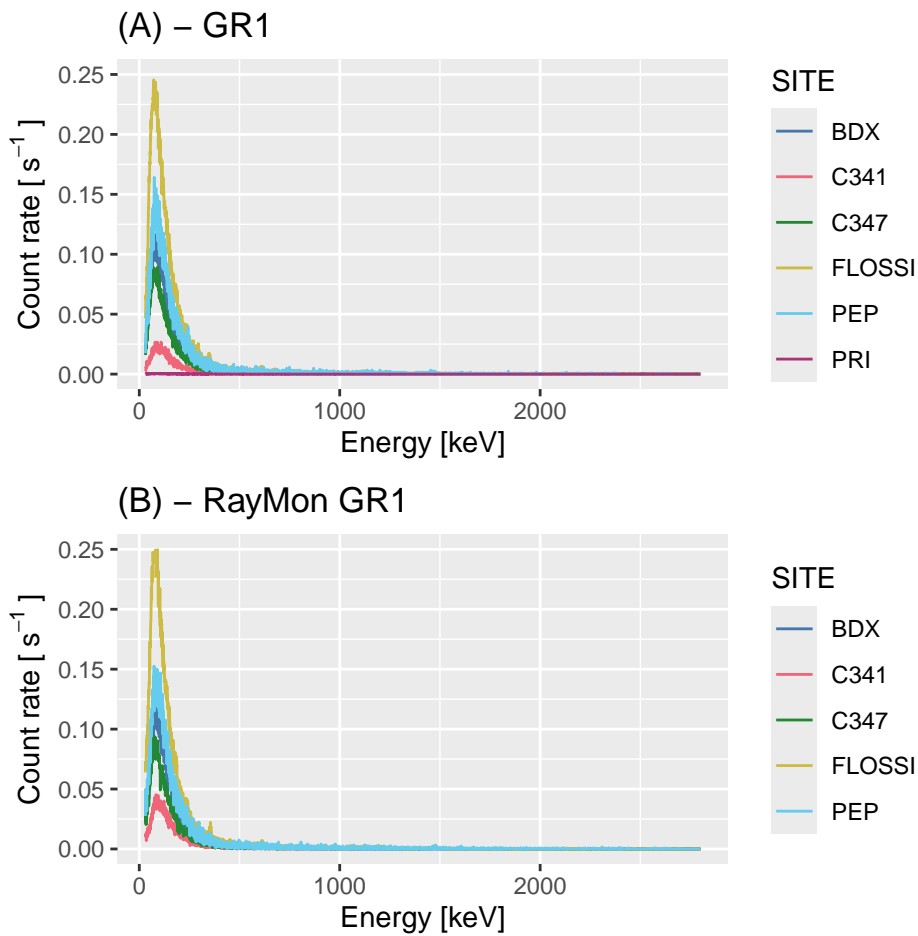

**Figure 6.** Channel/energy calibrated spectra as recorded in the reference sites as count rate against energy. (A) Spectra measured with detector GR1, (B) spectra measured with detector RayMon GR1. The spectra for both detectors are virtually identical in terms of peak position and count rates. The count rate for spectra C341 is significantly lower in (A) compared to (B). This is due to a software error (see maintext) and therefore this spectrum was discarded for later analysis. 'PRI' refers to the background spectrum recorded in the lead castle. We limited the x-axis to energy range later used for the integration: $30 - 2800 \, \mathrm{keV}$.

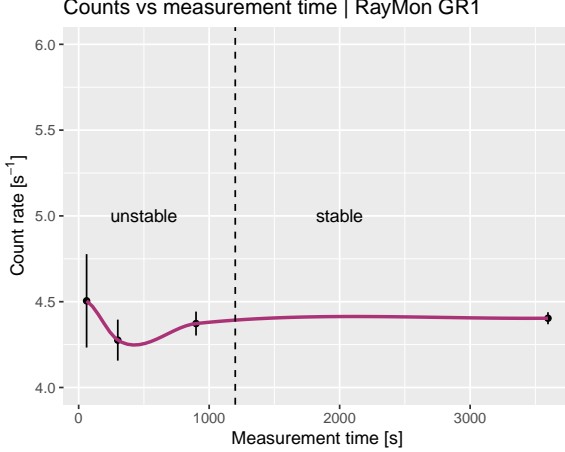

**Figure 7.** Sum of counts normalised to the measurement time recorded in the brick block at Archéosciences Bordeaux. After $20\,\text{min}$ the average count rate does not change anymore within uncertainties. The plot scales depending on the settings of $\eta$ (the energy threshold) and it was arbitrarily set to $200\,\text{keV}$ for this graph.

## 3.4 Minimum required measurement time

When we performed our measurements at the reference sites, we still needed more practical experience with the two detectors. Therefore, we opted for measurement times longer than the typical setting for the $\text{LaBr}_3$ probes (ca $10\,\text{min}$). Unfortunately, hour-long measurements for one sampling spot are often impracticable, considerably reducing the practicability of in-field measurements.

To assess the reasonably required measurement time for recordings, defined as a stable count rate within uncertainties in the field, we placed the RayMon GR1 detector in the brick block at Bordeaux (site: BDX) and started measurements for $60\,\text{s}$, $300\,\text{s}$, $900\,\text{s}$, and $3600\,\text{s}$ (Fig. 7). Given the similarity of both detectors, we assume that this experiment will also be valid for the GR1. In the post-processing we integrated all spectrum counts for the experiment using the integration settings given below and normalised them to the measurement duration. The estimated mean count rate is a little bit erratic over the first $500\,\text{s}$ before smoothing out after $20\,\text{min}$ of measurement time. Additional measurement time increases the count rate only slightly. We therefore conclude that $20\,\text{min}$ suffice in typical environments to determine a reproducible signal. This time corresponds to a total number of 4370 counts.

## 3.5 Threshold definition

In Sec. 2.3, we outlined the concept for defining the optimal energy threshold ($\eta$) above which the count rate correlates with the absorbed dose, regardless of the nature of the emitter and the matrix composition. The threshold is, in essence, a function of particle interaction with the (CZT) detector. Løvborg and Kirkegaard (1974) estimated the energy threshold for their setup ($3 \times 3$ in NaI detector) at $500\,\text{keV}$, Murray et al. (1978) settled on $450\,\text{keV}$ for their $2\,\text{in}$ diameter NaI(Tl) probe. Mercier and

Falguères (2007) calculated a threshold of 320 keV for their $1.5 \times 1.5$ in NaI(Tl) probe, a value later largely confirmed by simulations by Guérin and Mercier (2011) (their threshold value: 296 keV). Also, Duval and Arnold (2013) reported comparable values for NaI and LaBr$_3$ detectors of the same sizes (LaBr$_3$)(Ce): 358 keV, NaI(Tl): 322 keV). Our unpublished observations, employing the threshold method, suggest that the threshold shifts towards higher energies for larger detectors of the same material. This phenomenon is likely attributed to the increased proportion of photons registered from $^{40}$K, relatively to photons from the U- and Th-series. To compensate for this larger contribution of $^{40}$K photons in the distribution, the threshold shifts towards higher energies to ensure that the total count rate is proportional to the absorbed dose. Given the small volume of our CZT detector, we would position the threshold in the low-energy portion of the spectrum, not exceeding the values reported in the literature.

### 3.5.1 GEANT4 simulations

Figure 8A-B exhibits the simulations results for the three different matrices. We show the relative standard deviation between the number of counts above the energy threshold recorded during the simulations of 1 Gy generated with spectra of the radionuclides of the U-series, Th-series and the $^{40}$K. This standard deviation is minimized when the number of counts/energy above the threshold is less dependent on the radionuclide of a chain of origin of the $\gamma$-photons, i.e. when the number of counts above the threshold is proportional to the dose absorbed by the detector; despite of the origin of natural $\gamma$-rays. The minimum standard deviation, obtained between 192.5 keV and 242.5 keV for the count integration technique and 97.5 keV and 222.5 keV for the energy integration technique, correspond to the curves in Figs. 8A-B falling below 10 % of the relative standard deviation (horizontal dashed line in Figs. 8A-B). This represents the optimal energy range for setting the energy threshold for the detector $\eta_{sim}$ according to our simulations for the two integration techniques respectively.

### 3.5.2 Field measurements

### 3.5.3 Classical method

The "classical" method to determine $\eta_{exp}$ experimentally, are measurements in environments with different and ideally pure radionuclide compositions. Here we tried to determine the energy threshold using the calibrations sites at hand for the counts and the energy integration technique (Figs. 8C-D). The threshold is defined as the smallest relative standard deviation of all spectra normalised to the respective environmental $\gamma$-dose rate of the sites. In Figs. 8C-D we only show the results of the detector RayMon GR1.

For the count and energy integration technique, we obtained $\eta_{exp}$ at 99 keV and at 59 keV, respectively. Both values are considerably smaller than the results from our simulation. We will show later that the simulated $\eta$ is likely more accurate than the experimentally derived one. We attribute the difference to the similarity of the measured sites and to the fact that, although $\dot{D}_\gamma$ varies for all sites, the U/Th ratio is rather similar, likely leading to an unrealistically low value of $\eta$.

### 3.5.4 Calibration curve fitting

As an alternative to the simulation and experimental quantification of $\eta_{exp}$, we experimented with a different approach. We calculated the $\gamma$-dose rate response curve for different energy windows using `gamma::dose_fit()`. Amongst other values, the function returns the mean square of the residuals (MSWD), which we can use to approximate the quality of fit of our regression model. Values lower or higher than 1 indicate a poor model adaptation. We defined the moving lower energy limit as $E_i$ ($i := \{30, ..., 1000\}$) and $E_{max}$ was set to $2800\,\text{keV}$ to avoid counts from cosmic-rays. Figures 8E-F show the outcome of this calculation for both detectors (blue: GR1, red: RayMon GR1). Although the curve of the mean residuals differ, the divergence of the determined thresholds are small and we believe that this deviation is caused by the discarded data point C341 for GR1, which is the lower point in the calibration curve.

For the count *calculation* technique (not shown in Fig. 8E), the minimum in the search window between $30\,\text{keV}$ and $350\,\text{keV}$ was found at $91\,\text{keV}$. For the energy counting *calculation* technique, we located the value at $71\,\text{keV}$. Also these values are smaller than their simulated equivalents and the approach cannot compensate for the lack of differences between the measured sites. We therefore decided to continue with the simulated energy threshold values (i.e. $\eta := \eta_{sim}$).

### 3.6 Dose-rate calibration

With the threshold $\eta$ derived from $\eta_{sim}$, we can obtain our dose-rate model again with the function `gamma::dose_fit()`, but this time for a fixed count/energy threshold at $99\,\text{keV}$ and $59\,\text{keV}$, respectively. The results are shown in Fig. 9 for the detector GR1 (Fig. 9A) and RayMon GR1 (Fig. 9B). Visual inspection confirms a good fit of the model to the data. However, the calibration curves differ slightly between the two detectors, which is likely due to the lower number of available data points. The fitting parameters of both regression lines overlap with uncertainties (standard error as sum if weighted deviation from the fit; see York et al. (2004)). Acknowledging minimal variations between the CZT crystals and differences in housing and electronic, a perfect match is, however, not expected.

The package 'gamma' automatically fits the data for the energy threshold *calculation* technique and the counting threshold *calculation* technique for a given $\eta$. In Fig. 9 we have shown only the latter. However, both values are accessible and saved in the file `CAL_heiLUM_V0.rda` we made accessible at Zenodo (Kreutzer et al., 2024). As a reminder, we converted the radionuclide concentrations from the reference sites to dose rates using conversion factors compiled by Cresswell et al. (2018). These values influence the slope and intercept of the calibration curves. Because it may be desirable to apply additional calibrations based on other available conversion factors we repeated the calibration using conversion factors from Adamiec and Aitken (1998), Guérin et al. (2011), and Liritzis et al. (2013) (see data on Zenodo: (Kreutzer et al., 2024)).

### 3.7 Cross-check against natural site

The measurements at the Weiße-Hohl confirm once more that the two detectors exhibit very similar characteristics in terms of count rate efficiency (Fig. 10). Differences seem stochastic without visible systematic diversion over the measurement duration of $20\,\text{min}$. To estimate the uncertainties, we implemented a new routine in the 'gamma' R package (argument:

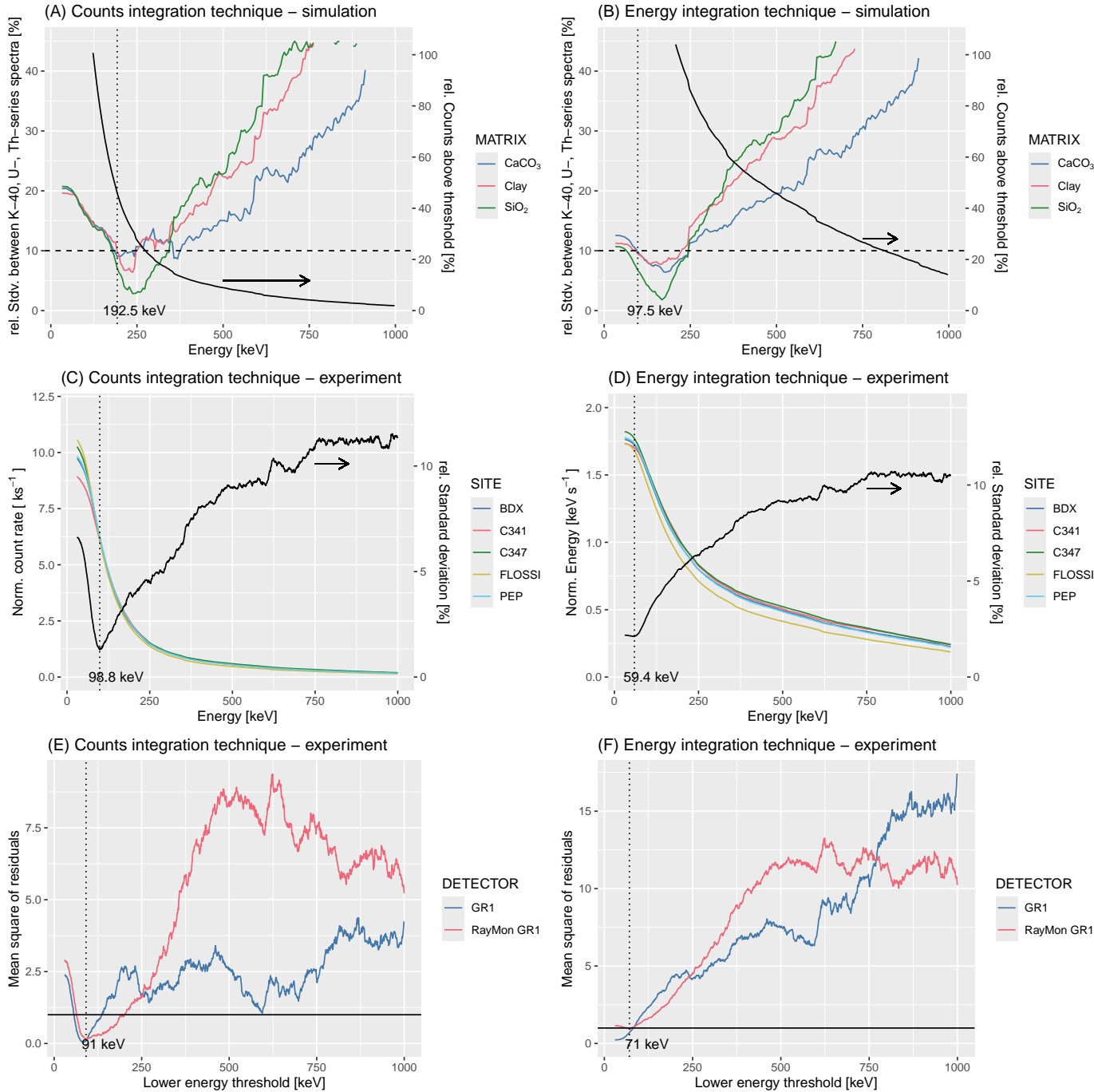

**Figure 8.** (A)-(B) *GEANT4* simulation results. (C)-(D) cummulative γ-ray spectra for all natural calibration sites normalised to the respective environmental γ-dose rate. Data only shown for RayMon GR1. (E)-(F) A variant of the experimental data as mean square of residuals (MSWD) of the dose-rate model fitting against the chosen minimum energy threshold. The values at 1 would indicate the best fit. The calculated energy threshold (η) is indicated in each of the plots as dotted line. Please keep in mind that while the three sets of graphs aim at showing the energy threshold using different methods, they neither represent the same data nor the same fitting method. For more details see main text.

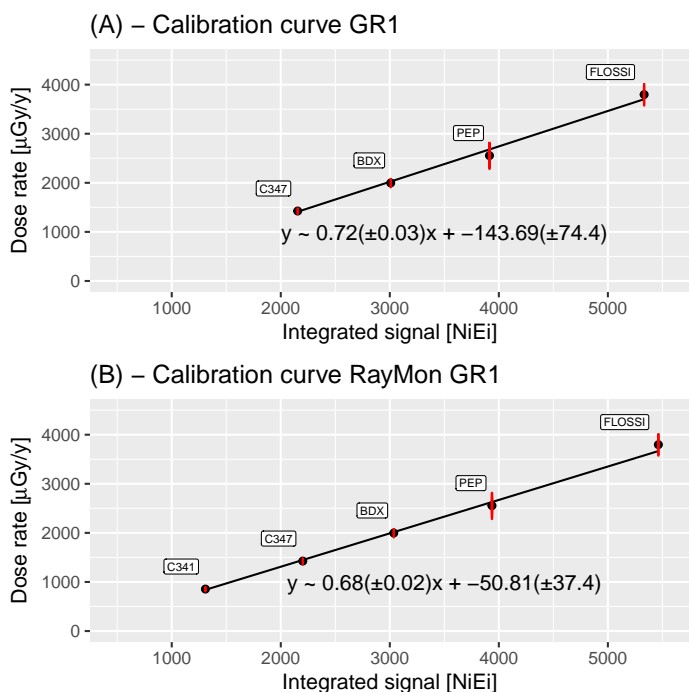

**Figure 9.** Dose-rate calibration curves for detector GR1 (A) and RayMon GR1 (B). Uncertainties are given as standard errors (for details see York et al., 2004). Shown is the known $\gamma$-dose rate from the reference sites against the integrated energy signal between the threshold $\eta$ (in keV) and 2800 keV. Note: The graphes give the impression that the uncertainties for the sites measured in France increase proportional to the size of the known $\gamma$-dose rates. This is a coincidence for our data subset and the effect not a real (cf. Miallier et al. (2009)).

`dose_predict(..., use_MC = TRUE)` that uses a Monte Carlo simulation approach, re-sampling from distributions for slope, intercept, and the signal to predict the dose rate on the regression line. We found that with this method, the uncertainties increases over the analytical approach, however, it should reflect the true uncertainties more realistically.

355     The water content from the sample site (sample code: WH2024) was estimated at 2.1 % in the laboratory and this value was used to correct $\dot{D}_\gamma$. Table 2 summarises the derived dose rate results for the two threshold integration techniques. $\dot{D}_{\gamma-final}$ is the arithmetic average of the values of these two techniques. The results for GR1 and RayMon GR1 agree within $2\sigma$ uncertainties. This observation is likely caused by the calibration of GR1 sitting on fewer data points. The comparison of $\dot{D}_\gamma$ against values derived from the radionuclide concentrations on WH2024 demonstrate a good agreement with field measurement

360     uncertainties. If we compare the CZT results (GR1 and RayMon GR1) with the laboratory derived $\dot{D}_\gamma$, both summarised as central values (e.g., Galbraith and Roberts, 2012) we obtained $1107 \pm 65\,\mu\mathrm{Gy}\,\mathrm{a}^{-1}$ (CZT) and $1105 \pm 11\,\mu\mathrm{Gy}\,\mathrm{a}^{-1}$ (laboratory).

    The results indicate a good homogeneity of the site reflected in the agreement between the field and the sampling dose rate. To get a better feeling for the sensitivity of $\dot{D}_\gamma$ as a function of $\eta$ for our detectors, we can calculate $\dot{D}_\gamma$ of WH2024 for different values $\eta$. For this experiment on the energy calibrated spectra, we have to repeat the dose-rate calibration curve fitting

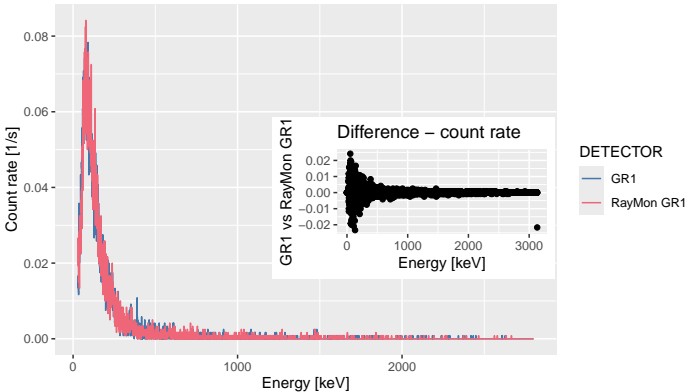

**Figure 10.** Gamma spectra recorded at Weiße Hohl (Germany) with the two detectors. The inset shows that the absolute count rate difference between the two detectors is randomly distributed.

**Table 2.** Dry $\gamma$-dose rate results for the sample WH2024 obtained with different methods. Uncertainties on the final dose rate are quoted in $2\sigma$ approximating the 95 % confidence interval. The dose rates for the CZT estimates include an systematic error contribution of 1% from the energy calibration and was corrected for the in situ water content. Beyond digits listed here, we calculated with the full precision as returned by the measurement devices. The CZT measurements are in situ measurements, the $\mu$Dose devices sampled 3 g of material each and the $\gamma$-ray spectrometer measurement used 88.3 g. REF_Central is the central value and its uncertainty from the laboratory derived $\dot{D}_\gamma$ values.

| DETECTOR | U | $\sigma_U$ | Th | $\sigma_{Th}$ | K | $\sigma_K$ | $\dot{D}_{\gamma-Ni}$ | $\sigma_{\dot{D}_{\gamma-Ni}}$ | $\dot{D}_{\gamma-NiEi}$ | $\sigma_{\dot{D}_{\gamma-NiEi}}$ | $\dot{D}_{\gamma-final}$ | $\sigma_{\dot{D}_{\gamma-final}}$ |
|---|---|---|---|---|---|---|---|---|---|---|---|---|
| GR1 | NA | NA | NA | NA | NA | NA | 1095.35 | 186.33 | 1063.28 | 170.76 | 1105.15 | 258.54 |
| RayMon GR1 | NA | NA | NA | NA | NA | NA | 1084.00 | 109.80 | 1079.84 | 92.34 | 1107.82 | 146.86 |
| $\mu$Dose+ (05) | 2.58 | 0.37 | 10.91 | 0.89 | 1.18 | 0.03 | NA | NA | NA | NA | 1115.65 | 36.27 |
| $\mu$Dose (25) | 3.14 | 0.36 | 9.72 | 0.84 | 1.23 | 0.06 | NA | NA | NA | NA | 1132.54 | 34.25 |
| HPGe Prisna | 3.10 | 0.02 | 8.70 | 0.07 | 1.20 | 0.01 | NA | NA | NA | NA | 1069.03 | 2.92 |
| REF_Central | 3.09 | 0.02 | 9.44 | 0.53 | 1.20 | 0.01 | NA | NA | NA | NA | 1105.30 | 21.25 |

*Note:*

U, Th concentrations in $\mu$g g$^{-1}$, K in % | dose rates in $\mu$Gy a$^{-1}$. | Prisna is a $\gamma$-ray spectrometer in the Archéosciences Bordeaux laboratory

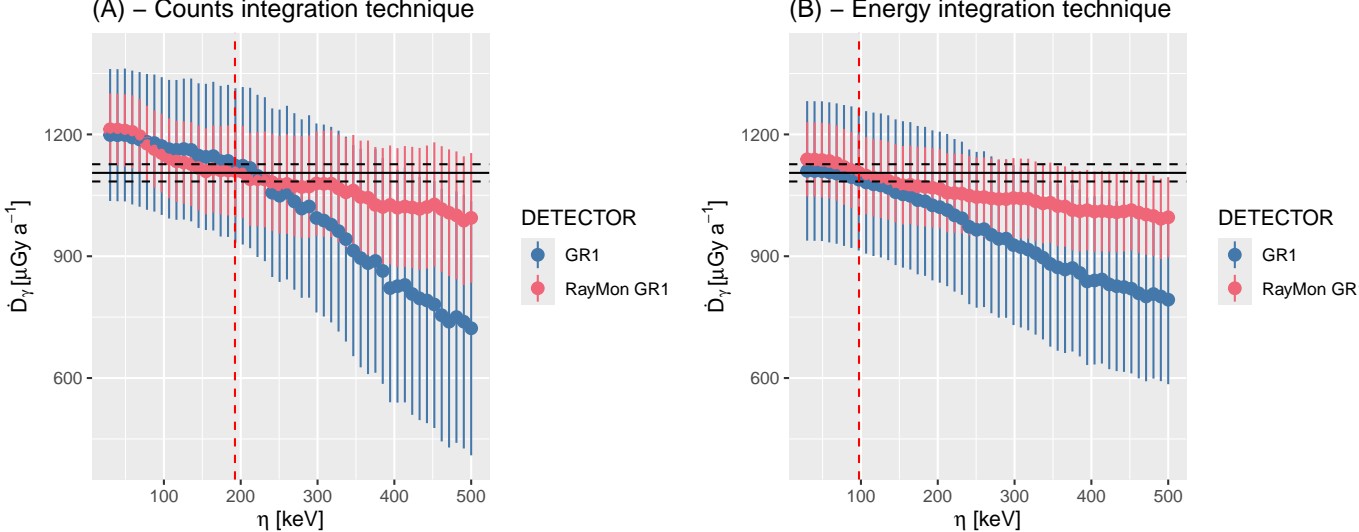

**Figure 11.** Estimations for $\dot{D}_\gamma$ for different values of $\eta$ for the two investigated detectors. The solid line shows the $\dot{D}_\gamma$ of the measured site Weiße Hohl derived from laboratory radionuclide estimations and the dashed lines its uncertainties. All uncertainties are shown as $2\sigma$ values. For further details see main text.

and then predict the new dose rate given the newly derived calibration for $\eta_i$ (in keV) with $i := \{30, 40, 50, ..., 530\}$. The upper integration energy was set to $2800\,\text{keV}$ analogue to the previous calculations.

We assume that the laboratory-derived $\dot{D}_\gamma$ are the benchmark value we want reproduce. Figure 11 offers insight in the evolution of $\dot{D}_\gamma$ for GR1 (blue) and RayMon GR1 (red). The solid line is central value reference for WH2024 from laboratory measurements, and the dashed lines show the $2\sigma$ uncertainties (calculation after the Central Dose Model; Galbraith and Roberts (2012)). Setting aside of the calibration-caused discrepancy between the two detectors, both detectors perform similarly between a threshold of ca $30\,\text{keV}$ and $200\,\text{keV}$ for the counts integration technique and $30\,\text{keV}$ to $150\,\text{keV}$ for the energy integration technique and overlap before falling systematically below the laboratory-derived reference value. The uncertainties of all analyses overlap until a threshold of ca $300\,\text{keV}$ and it appears that for our setting and tested environment, the output is relatively insensitive to the exact value picked for $\eta$ (given the uncertainties). The picked values, however, seem to nearly ideally reproduce the benchmark value, while lower values for $\eta$ as indicated in our experiments would likely overestimate the true $\dot{D}_\gamma$.

## 4   Discussion

Our study reported the performance and dose-rate calibration procedure of two portable semiconductor-based portable $\gamma$-ray spectrometers. Both devices host a similar CZT detector that can be operated at ambient temperature, i.e., in situations typical for environmental dose rate measurements as part of trapped-charge dating studies. Unlike literature reporting on $\gamma$-ray

measurements in the field that used NaI(Tl) or LaBr$_3$ probes with inch-size diameters, our detectors are considerably smaller (crystal volume $1 \, \text{cm}^3$), and the systems have a low power consumption, boosting their appeal for trapped-charge dating studies despite that no previous experience was available addressing our field of application. This seems surprising, given the body of available literature about CZT detectors. However, usually, those studies aim at nuclear radiation monitoring (e.g., Alam et al., 2021) or identifying artificial radio-nuclides in environmental studies (e.g., Rahman et al., 2013) for which such detectors are primarily designed.

On the plus side, this feature of the detectors simplified the energy/channel calibration with artificial radionuclides because of the detectors' sensitivity to those nuclides. Our energy calibration exhibited peak positions in excellent agreement for both detectors and we concluded that we could apply one single energy calibration. This approach was valid for us, but other detectors likely require separate channel/energy calibration. Although we did not observe a shift in the channel/energy calibration with temperature during all experiments, we highly recommend an energy/channel calibration as part of the post-processing because all subsequent analyses depend on it.

Dating studies require an accurate reading of a sediment matrix's natural $\gamma$-radiation field of unknown radionuclide composition in a $4\pi$ geometry at the sampling position. Our study proved that both detectors can achieve this in a reasonable time of $20 \, \text{min}$. This value likely works for many environments typically encountered in trapped-charge dating applications. Still, it might be too short for accurate dose rate estimations in settings with a low amount of natural radionuclides or if higher precision is desired. Hence, in case of doubt, measurement times should be adjusted. We recommend a minimum measurement duration of $60 \, \text{min}$ to obtain the dose-rate calibration curve with a good counting statistics.

A crucial part of our contribution was the determination of the energy threshold $\eta$ above which the count/energy rate is proportional to the dose rate for natural radioactive elements. Given the highly comparable performance characteristics of both detectors, our results can be easily used by others with the same type of detector without repeating all experiments. We tested three different methods (simulation, classical measurements, dose-rate response curve fitting) to determine this threshold and opted for the results from the simulation since the measured natural sites were likely not diverse enough. Because we had access to general schematics provided by the manufacturer, we also believe that the *GEANT4* simulation should be fairly accurate. The cross-check to the natural loess site indicates that the assumption made to simplify the simulation had no significant impact on the results. The threshold found here is considerably lower than results obtained in studies with NaI(Tl) or LaBr$_3$ probes that place $\eta$ at ca $300 \, \text{keV}$ or higher (Mercier and Falguères, 2007; Guérin and Mercier, 2011; Duval and Arnold, 2013). This balances to some extent the lower absolute efficiency of the tiny CZT detectors because it allows exploiting a larger portion of the recorded spectrum.

Nonetheless, it should be kept in mind that all three approaches, simulation, classical experiments, and dose-rate curve fitting, have different meanings. Under the assumption of correct input parameters, the simulation investigates the interaction of the $\gamma$-photons with matter for different scenarios and can hence truly determine a range above which the threshold assumption is valid. In other words, the simulation results have merit and provide a solid basis for setting of the thresholds. On the contrary, the experiment findings depend on the matrix composition of the host rock, which in our case, is very similar if translated into relative $\gamma$-dose rate contributions from the different radionuclides. We did not observe matching pattern for the threshold from

the measurements and the simulation and without simulation, a meaningful determination of $\eta$ still requires measurements of emitters with pure radionuclide composition, such as the Oxford blocks (Rhodes and Schwenninger, 2007).

The threshold quantified in our study is likely not much different for detectors of similar size and with a comparable CZT detector. Therefore we argue that the threshold settings can be adapted if a simulation or a measurement is not possible. This suggestion is further supported by our tests of the Weiße-Hohl measurement with shifting thresholds and minor difference in the detection characteristics will not bias the outcome for the $\gamma$-dose rate. Future work should investigate the calibration curve at very low dose rates and in very different environmental settings, since this was not tested in our study.

The $\dot{D}_\gamma$ results of the Weiße-Hohl reflect mainly statistical variations of the different analytical methods. Striking but not puzzling is relatively large coefficient of variation ($c_\nu$) of portable CZT detector results (GR1: 11.9%; RayMon GR1: 6.8%) compared to the laboratory measurements (ca 1.7% or lower). Duval and Arnold (2013) reported $c_\nu$s around 5% comparable to our laboratory measurements and results of LaBr$_3$ measurements calibrated at the Clermont-Ferrand sites with more data points typically yielded a $c_\nu$s of 5% or better (e.g., Kreutzer et al., 2018a, their Table S9) (typical values: live time: ca 600 s; integrated counts: ca 24000 counts; count rate: $45\,\mathrm{s}^{-1}$)

Furthermore, the larger uncertainties of GR1 compared to RayMon GR1, seem to diminish the overall good performance. For GR1, the weaker performance (larger uncertainties) results from only having three calibration points available, which would disappear with an additional point. More generally, we argue that this precision can be significantly improved with more points for the dose-rate calibration curve. Those points can be added at any time later, for instance, by measuring more sites around Clermont-Ferrand (Miallier et al., 2009). In such case, however, a check on the energy calibration is mandatory before and after going to the field to monitor potential shifts of the energy spectrum that might be caused by temperature or other technical reasons. Although our results did not encounter such shift, all experiments were carried out in a very short time window.

Finally, what we did not expect of these CZT detectors but should be mention for completeness is that our findings show that those detectors are unsuited for applying the "window" method in environments and for measurements durations typical for our field of application. This would require hour-long measurements to achieve acceptable error margins. For the determination of radionuclide composition laboratory based analytical techniques are unmatched in their effectiveness and precision and they also allow to derive $\alpha$ and $\beta$-dose rate components.

## 5  Conclusions

The primary aim of our study was to test and evaluate the performance of two portable CZT detectors for deployment as active in situ detectors in trapped-charge dating applications. To that end, we measured spectra on natural reference sites with known radionuclide composition in France to derive a dose-rate calibration curves for our two detectors. Background measurements in a low-radiation setting exhibited negligible count rates that can be ignored.

To determine the optimal energy threshold above which the matrix composition of the measured site does not bias the integrated signal to $\gamma$-dose rate relation, we performed energy-matter interaction simulations using *GEANT4*. The simulation indicated a suitable energy threshold between 192 keV and 242.5 keV for the counts integration technique and 98 keV and

222 keV for the energy integration technique. We compared those with thresholds derived from cumulative spectra and signal-dose rate regression lines for the two different integration techniques and we found a value of 91 keV for the counting threshold integration and 71 keV for energy counting integration technique. However, given the results from the reference loess site with know radionuclide composition, we discarded the experimentally derived energy thresholds as they are likely too low because of the high similarity of the investigated natural sites. To record a $\gamma$-dose rate in typical natural sediment environments, we recommend a measurement time of at least $20\,\mathrm{min}$ (this approximates to a total of 4500 counts or better).

A check of our results through measurements at the homogeneous loess deposit near Heidelberg for which we derived the radionuclide composition in the laboratory, confirmed an excellent match of field and laboratory methods, however, with considerably larger (but perhaps more realistic) uncertainties for results from the CZT detectors. Finally, we argue that refined calibrations can further reduce those uncertainties on more sites. Future work may want to extend our calibration curves and explore the performance of the detector in more extreme (low and high) natural radiation fields.

*Code and data availability.*  Raw and partly processed data and R code used in this study is available on Zenodo (https://doi.org/10.5281/zenodo.13731839, last access: 2024-09-10).

*Author contributions.*  SK: Writing – original draft, Validation, Methodology, Formal analysis, Conceptualization, Funding acquisition, Data curation, Software. LM: Writing – review and editing, Methodology, Formal analysis. DM: Writing – review and editing, Resources. NM: Writing – review and editing, Methodology, Formal analysis, Conceptualization.

*Competing interests.*  We declare no conflict of interest and did not receive funds other than what was stated. The Heidelberg Luminescence Laboratory procured the tested systems with its own funds. We contacted the equipment manufacturer to obtain technical information required for the dose-rate simulation and regarding the temperature stability. Other than that, we did not liase with the manufacturer.

*Disclaimer.*  Our contribution aims to provide information on the practical use of systems available commercially off-shelf to the trapped-charge dating community and report calibration and performance results. However, we do not favour a particular supplier, and our manuscript must not be understood as a purchase recommendation.

*Acknowledgements.*  We thank Nicolas Frerebeau for his very responsive support as maintainer of the R package 'gamma'. We are grateful to Maryam Heydari for performing the measurements at the Weiße Hohl. We further thank Michael Faske from the Scientific Workshop Service of Heidelberg University for designing and printing the strain relief adapter for the GR1 detector and agreeing to share it under CC

BY-NC licence conditions. Dennis Gross and Jochen Schreiner enabled the access to FLOSSI and organised the removal of the tree in front

475   of the block. Martin Autzen and an anonymous reviewer provided as with helpful comments and suggestions.

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
