# Peer review of "Environmental Gamma Dose Rate Measurements using CZT Detectors"

_Geochronology, 2024_

## Author Response (AR1)

As required by the journal system, we herewith repeat our previous response to the reviewers. Our responses were already logged in the open discussion. We have revised our manuscript as detailed and we thank the Geochronology editorial team for their support.

**Reviewer 1**

We thank the reviewer for the comments and suggestions. We have revised our text accordingly and will upload a modified version of the manuscript including the changes listed below.

> My other concern goes to the GEANT4 simulation, particularly its low-energy aspects. A more detailed discussion of the geometry, materials, and limitations would be beneficial.

Generally, the GEANT4 simulation accounts for low-energy aspects and the detector geometry is based on sketches provided to us by the manufacturer including information regarding the materials. We provide the code used for the simulation on Zenodo to avoid any ambiguity. To accommodate the request for additional details in the main text, we will include further information.

> Finally, I would advise to round values according to their uncertainty, there are generally accepted rules.

Thank you for pointing this out. This comment likely refers to Table 2: We had decided to use the uncertainty of the HP-Ge detector as benchmark and therefore we show two digits in the table consistently for all methods. However, for the final results we will remove the digit as this numbers is indeed not meaningful given precision of our approach.

> L169-170: I agree that the dead time for environmental radioactivities should not influence the results. However, the sensitivity change with the energy might be a significant factor. Could you elaborate more why it can be neglected or what precisely you refer to when referring to sensitivity?

This is indeed a valuable comment because our writing obviously caused misunderstandings. What we meant was not the simulation but potential photon-energy related detection sensitivities of the detector system. Our GEANT4 modelling accounts for different photon energies but we wanted to indicate that there might be other effects we simply ignore. However, after an internal discussion with decided to rephrase this part of manuscript because it sits on speculation on our part and invites to misunderstandings. The new phrase will read:

"We did not consider dead times because we assume that this phenomenon has a low impact on determining the count/energy threshold."

> L297-305: I was wondering how accurate is the low energy part of the simulation. How well are mimicked detector enclosure, was the material and it thickness available? Is the detector response linear in tens of keV part?

We wrote in the figure caption of Fig. 2. "Information kindly provided by the Kromek Group plc". We will update the figure caption to details that we had access to those information (except for details of the electronic). Therefore we believe that the simulations are fairly accurate.

**Minor comments**

We addressed all minor comments as suggested, here the record to fhe applied changes as documentation.

> L23: Asserting or assessing?

Corrected to assessing; thank you!

> L42: Appeal?

We replaced this by "are indicated"

> L45: I would use "statistically agree" instead of "agree within uncertainties". Usually expanded uncertainty is not used.

Done.

> L79-82: Those sentences do not make sense when they are together.

Indeed, thank you for pointing this out. We separated the sentences better.

> L213: "..(2019) ." please remove space

Done.

> Figure 7. Please set y axis from 4 to 6 cps.

Done.

> L281-284: Is there instability above what would be expected from Poisson distribution? If there is no such instability detected would not use term stable unstable words. They usually refer to phenomenon that arise from the detector system itself not from counting statistic.

Thank you for pointing this out. The effect is purely stochastic and not related to any instability of the measurement system. We will rephrase the sentence.

> L292-295: Cold you explain the reasoning behind this rule.

We will rephrase the sentence as follows to provide a better explanation:

"Our unpublished observations, employing the threshold method, suggest that the threshold shifts towards higher energies for larger detectors of the same material. This phenomenon is likely attributed to the increased proportion number of photons registered from $^{40}$K, relatively to photons from the U- and Th-series. To compensate for this larger contribution distribution of $^{40}$KK photons, the threshold shifts towards higher energies to and ensures that the total count rate is proportional to the absorbed dose. Given the small volume of our CZT detector, we would position the threshold in the low-energy portion of the spectrum, not exceeding the values reported in the literature.:

> L314: "We wil l later show"

Corrected, thank you.

> L326: Fig. 8E - missing dot

Corrected.

> L329: missing dot

Done.

> L332: missing space "99keV"

The line includes a half-space; which is correct.

> L336: CZY?

Corrected.

> L344: "... (see data on Zenodo: (Kreutzer et al., 2024))." - missing ")"

Done.

> Fig. 2 - 11. Could you unify as much as reasonably possible the font size?

Thank you for pointing this out. However, all figures were already produced consistently "print ready" for a two column layout. In the single-column layout required by Copernicus for the manuscript, the used **\textwidth** command however, scales the figures differently. This will look better in the production process, if the manuscript becomes accepted. Still, depending on the final decision of the typesetter about the figure placement, we may have to scale the fonts again.

> Fig. 2 - 11. Could you use it systematically: Title Case, Sentence case or lowercase.

Thank you, we corrected the writing.

> Fig. 2. How big was the area taken for the MC simulation?

The area is 2.65 m^2, the volume 4.4 m^3. We will add those information to the figure captions.

> Fig. 8. On all subplots it now is difficult to assess what is on the left and right axis.

Thank you, we will add arrows to each of the four figures to indicate the corresponding scale.

> CaCO3 (subscript)

We will correct the figure.

> Fig. 11. The uncertainties might be somewhat misleading. From my understanding the error bars have the same component and therefore for other measurements I am wondering about the interpretation.

The error components are governed by the calibration and will not become much smaller for other measurements unless more sites are used for the calibration. This part is indeed systematic, because it applies to all measurements, the random part is negligible. What we wanted to show here is that we cover the expected dose rate range even for a non-optimal threshold. In other words, even the threshold was not estimated that carefully, within uncertainties the results will be correct and this is the limitation of the method.

**Reviewer 2 (Martin Autzen)**

Dear Martin,

Thank you very much for your positive feedback.

> I would like to see the section on Geant4 simulation expanded to include more details of the setup. Currently it is not clear if the detector is simulated to be at the center of the sediment matrix or at a distance of 160 cm. The dimensions of the sediment matrix is also not given. Figure 2 also appears to be in conflict with the description of the setup or maybe I am misinterpreting it. What production cut was used? 1 keV?

I agree with this comment and the limited information provided was also flagged by reviewer #1. Although the simulation code is included in the data provided alongside our manuscript, I acknowledge that this is not an entirely straightforward matter and can be done better and more reader friendly.

Consequently, I believe that essential information should be integrated into the main text and will be added. Besides, to answer the questions: The sediment box had dimensions of (1.635 × 1.635 × 1.67) m, which, when combined with the dimensions of the detector (160 cm), resulted in a total volume of 160 cm^3. Furthermore, the production cut was set to 0.5 keV. However, I will verify this information with Loïc before we re-submit the updated manuscript.

> L15: limited variability

Thanks for spotting this!

> L31: consider removing commas in "both, laboratory and field, measurements"

Fixed and slightly rephrased to improve the reading.

> L60: Gamma spectrometers need an export license due to being dual use and cannot be brought into every country, consider adding to sentence.

Thanks, this is indeed an issue to consider and we will add it.

> L80-82: full stops could be replaced by commas to improve reading flow

We will not change this for the moment, because in the production process step usually the copyeditor will add a lot of language changes.

> L130: emitted/source instead of radiator?

Modified.

> L301: "despite of the origin of the y-rays of natural origin." Is the first origin where they originate in the matrix?

Thank you for spotting this; corrected.

> L336: CZT instead of CZY

Corrected.

> L351: uncertatines of the already implemented analytical approach increases?

Thanks, we rephrase this part.